# Evolution of Defects in CVD-W Irradiated by H/He Neutral Beam Using Positron Annihilation Spectroscopy

Xuefen Tian [1], Xiang Liu [2], Min Gong [1], Weidi He [1], Xinge Fu [1] and Aihong Deng [1,*]

[1] College of Physics, Sichuan University, Chengdu 610064, China; 18382204138@163.com (X.T.); mgong@scu.edu.cn (M.G.); baiyeqiandi@outlook.com (W.H.); aaaafin@163.com (X.F.)

[2] Southwestern Institute of Physics, Chengdu 610041, China; xliu@swip.ac.cn

[*] Correspondence: ahdeng@scu.edu.cn; Tel.: +86-152-0838-0562

**Abstract:** One of the key problems for the application of nuclear fusion energy is to select the suitable plasma facing materials (PFMs). Among the W-based materials, CVD-W exhibits some unique advantages. In order to estimate the performance of CVD-W under the fusion environment, the vacancy-type defects and their evolution are investigated by the Doppler-broadening slow positron beam analysis (DB-SPBA) combined with SEM (scanning electron microscope). There are two kinds of neutral beam irradiation, the pure H neutral beam and the H + 6 at.% He neutral beam irradiation, which are performed at the neutral beam facility GLADIS (IPP, Germany). The surface temperatures of CVD-W irradiated by H (H + 6 at.% He) are 850 and 1000 (700 and 800 °C). By comparing the samples under different conditions, the defect evolution of CVD-W is obtained. As for the pure H neutral beam irradiated samples, the DB-SPBA results demonstrate that the CVD-W sample at the surface temperature of 1000 °C, compared to the 850 °C sample, shows a decrease in $S$ parameters, which is due to the reduction of vacancy-type defect concentration. The defect damage layer in 1000 °C sample is narrower than that of 850 °C sample and the defect type tends to be consistent in 1000 °C sample. The SEM results suggest that the surface damage of the 1000 °C sample was recovered to some extent. As for the H + 6 at.% He neutral beam irradiated samples, compared with the CVD-W sample at the surface temperature of 700 °C, the 800 °C sample shows an increased $S$ parameters, which can be attributed to the volume increase of vacancy-type defect. The defect damage layer in the 800 °C sample is wider than that of the 700 °C sample. Both the H + 6 at.% He irradiated samples show complex defect types. The surface of the 800 °C sample exhibits more dense pinhole damage structures compared to that of the 700 °C sample.

**Keywords:** CVD-W; H/He; DB-SPBA



## 1. Introduction

Nuclear fusion energy has been widely concerned and highly valued for its safe, clean, and environment-friendly traits. However, there are still a series of problems during the process of commercialization and engineering application of nuclear fusion energy, a key one of which is to choose the appropriate plasma-facing materials (PFMs) [1]. So far, W (tungsten) has been considered as the most promising candidate material for the first wall and divertor material in the international experimental fusion reactor (ITER) and demonstration fusion reactor (DEMO) due to its high melting point, high thermal conductivity, low sputtering rate for light elements, and low activation after neutron irradiation [2,3]. In the operation of fusion reactor, the PFMs are subjected to extremely severe conditions: the high flux plasma particle bombardment, high energy neutron irradiation, steady-state thermal shock, and transient thermal shock [1,4]. There are some drawbacks that limit the application of W [5–7] under the fusion environment, including the low temperature embrittlement (high ductile-to-brittle transition temperature DBTT > 400 °C), the recrystallization embrittlement (recrystallization temperature is about 1200 °C), and the irradiation embrittlement.

In order to improve the properties of W materials, some methods have been developed, such as the alloying [8,9], the oxide dispersion strengthening [10], and the carbide dispersion strengthening techniques [7,11,12]. However, the W-based materials produced by the above methods still exhibit poor machinability, purity and density. Fortunately, among the W-based materials, the W films grown by chemical vapor deposition (CVD) technique, called CVD-W, have some advantages, which are high purity, good thermal conductivity, and high heat load [13–15]. CVD-W also has the merits in the production of the heterogeneous parts with complex shapes and different sizes [16]. Generally, CVD-W is produced by the reaction between $H_2$ and $WF_6$. Different deposition rates have been obtained by adjusting the deposition conditions, and the fastest deposition rate at present can reach 1.0 mm·$h^{-1}$ [17–19]. Additionally, CVD-W has the good adhesion performance with the first wall and divertor assembly [17]. Moreover, the rapid deposition CVD-W has columnar crystal and pyramid structure characteristics [18,20], leading to the excellent performance of the CVD-W in the fusion operation [18,21,22]. Previous research showed that CVD-W had better thermal stability than that of the rolled W [13]. Compared with the forged W, the polished CVD-W has a higher cracking threshold up to 0.28–0.33 GW·$m^{-2}$ under disruption-like thermal shock and 0.33–0.44 GW·$m^{-2}$ under repetitive edge localized modes like thermal loadings, respectively [13,23]. Jia et al. [24] found that there were only a few bubbles on the surface of CVD-W sample exposed to the D plasma, which indicated that the CVD-W has good resistance to D plasma irradiation. Recently, Yi et al. [25] reported that the thermal diffusion coefficient of CVD-W after neutron irradiation was similar to that of the unirradiated rolled W. All the above studies imply that CVD-W will be an important candidate material in future fusion reactor.

Under operation, the divertor materials will be exposed to the H (D or T)/He mixed plasma with the flux of $10^{24}$–$10^{28}$ $m^{-2}$·$s^{-1}$. He is mainly produced by D-T reaction and its concentration in burning plasma can be 5–10% (atomic fraction) [26,27]. In addition, the divertor materials will also be subjected to a heat flux up to 10 MW·$m^{-2}$ [28,29]. The high energy neutral beam irradiation can cause some damage for the divertor components and PFMs. This issue has caused wide concern. Some publications consider PM-W, however, only a few focus on CVD-W. Liu et al. [26] reported that the surface morphology of CVD-W after pure H neutral beam irradiation was different from that after H + 6 at.% He neutral beam irradiation, illustrating the influence of mixing He on the surface morphology of the samples. In the paper, we use the DB-SPBA (Doppler-broadening slow positron beam analysis) to evaluate the defect evolution in CVD-W samples irradiated by H/He neutral beam on an atomic scale. DB-SPBA is an effective and commonly used method to study the evolution of vacancy-type defects in the metals. In order to better illustrate the problem, the SEM (scanning electron microscope) images are also given. The study is of great significance to understand the resistance of CVD-W samples to the H/He neutral beam irradiation and it also provides valuable help to estimate the actual applications of CVD-W in fusion reactors.

## 2. Experiment

### 2.1. Sample Preparation

The W coating samples in the work were made from Xiamen W Co., Ltd., China. The W coatings were produced by chemical vapor deposition (CVD) and obtained from the chemical reaction between $WF_6$ with the purity of 99.99 wt.% and $H_2$ with the purity of 99.999 wt.%. The CVD-W coatings were about 2.5 mm in thickness, deposited on a molybdenum plate with a thickness of 3.5 mm. The CVD-W coatings with a high purity larger than 99.9999% have a typical columnar crystal structure. The density of CVD-W is 99% that of bulk W, and it is 19.23 g/$cm^3$. The purity and density values of CVD-W are supplied by the manufactures (in Xiamen). The CVD-W samples were cut into pieces with the dimension of 20 mm × 12 mm × 6 mm. The Vickers hardness of CVD-W is 430 HV and its thermal conductivity coefficient is higher than 180 W/(m·K). The polishing was performed to get the roughness of CVD-W less than 0.5 μm, followed by annealing

at 1000 °C for 5 h in order to remove the residual stress caused by the preparation and polishing process.

### 2.2. H/He Neutral Beam Irradiation

The H/He neutral beam irradiation was performed on the high heat flux test facility GLADIS at Max-Planck-Institute Plasmaphysik, Germany [30]. There were two kinds of neutral beam irradiation conditions, the pure H neutral beam irradiation and the H + 6 at.% He neutral beam irradiation. The pure H neutral beam irradiation occurred with the heat loading of 10 MW/m$^2$ and the corresponding accelerating voltage was 29 keV. The total irradiation time of the pure H irradiation was 880 s and the total fluence amounted to 3.4 × 10$^{24}$ atoms/m$^2$. The values of H + 6 at.% He were 8 MW/m$^2$, 27 keV, 2100 s and 6.7 × 10$^{24}$ atoms/m$^2$, accordingly. On the basis of the Gaussian distribution of the beam flux and the dimensions of the mock-ups, the heat and particle flux values on the edge position samples were 85% of the samples' values at the middle position [26]. The CVD-W samples were brazed on a heat sink material of CuCrZr with the active water-cooling. A two-color pyrometer and an IR camera were used for monitoring the samples' surface temperature. The detailed irradiation information is listed in Table 1.

**Table 1.** Irradiation parameters of CVD-W samples.

| Sample No. | Neutral Beam | Heat Flux/MW·m$^{-2}$ | Fluence/m$^{-2}$ | Surface Temperature/°C |
|---|---|---|---|---|
| No.1 | H | 10 | 3.4 × 10$^{24}$ | 850 |
| No.2 | H | 10 | 3.4 × 10$^{24}$ | 1000 |
| No.3 | H +6% He | 8 | 6.7 × 10$^{24}$ | 700 |
| No.4 | H + 6% He | 8 | 6.7 × 10$^{24}$ | 800 |

### 2.3. Doppler Broadening Spectrometry (DBS)

Firstly, the samples after irradiation were directly put into SEM for surface morphology observation. Then, the DBS of positron annihilation measurements (at the Institute of High Energy Physics, Chinese Academy of Sciences) were carried out to study the evolution of vacancy-type defects in the CVD-W samples after irradiation. The $^{22}$Na radioactive source is selected as positron source. After the positrons (e$^+$) with a certain energy are implanted in the samples, they will eventually annihilate with the electron (e$^-$) during their diffusion process. Before e$^+$ annihilate with the e$^-$ they will lose their energy up to thermalization, and the energy for e$^+$ is ~0.025 eV and a few eV for e$^-$. The kinetic momentum of e$^+$ is negligible. Due to the kinetic momentum of annihilated e$^-$, the energy of annihilation ray is shifted by the Doppler broadening. The Doppler shift is given by:

$$\Delta E = \frac{cP_L}{2} \tag{1}$$

where $c$ is the speed of light and $P_L$ is the component of the kinetic momentum of the e$^+$ and e$^-$ annihilation pair along the propagation direction of photo. The DBS mainly reflects the kinetic momentum distribution of the e$^-$. The high purity germanium detector (HPGe) is used to record the two 511 keV photons produced by annihilation emission. Due to the poor energy resolution of DBS, the *S* and *W* parameters are often used to analyze the change of DBS [31]. The *S* (*W*) parameters reflect the momentum information of the low (high) momentum electrons, and they are related to the positron annihilation with the valence (core) electrons. If the positrons are trapped and annihilate at the vacancy-type defects in the samples, the *S* parameters will increase and the *W* parameters will decrease correspondingly. The *S* and *W* values at different depth in the sample can be obtained by changing the energy of the incident positrons. The positron incident energy varying

from 0.18 keV to 20.18 keV with 8 nm beam diameter was used in the paper. The average injection depth of the positrons is given by [32]:

$$\bar{Z} = AE^n/\rho \tag{2}$$

where $A$ and $n$ are constants ($A = 4 \times 10^{-6}$ g $\cdot$ cm$^{-2}$ $\cdot$ keV$^{-1.6}$, n $= 1.6$), $\rho$ (g/cm$^3$) is the density of material (for CVD-W, it is 19.23 g/cm$^3$) and $E$ (keV) is the incident positron energy. Because the positrons with a certain energy in the material can diffuse and have a certain of depth distribution [32], the *S-E* curves cannot directly reflect the depth distribution of the defects. Generally, the *S-E* curves (the *S* parameter as a function of the positron energy $E$) can be fit by the VEPFIT software [33], which can solve the one-dimensional diffusion equation of the positrons [32] that is given by:

$$D_+ \frac{d2}{dz^2} n(z) - \kappa_{eff}(z)n(z) + P(z, E) = 0 \tag{3}$$

where $n(z)$ represents the positron density at distance $z$ from the surface, $P(z, E)$ is the positron implantation profile at a given $E$, $D_+$ is the positron diffusion coefficient. The positron diffusion length ($L_+$) is related to the positron diffusion coefficient by:

$$L_+ = \left[ \frac{D_+}{\kappa_{eff}(z)} \right]^{1/2} \tag{4}$$

where the effective escape rate $\kappa_{eff}(z)$ is given by $\kappa_{eff}(z) = \lambda_b + \mu C_d(z)$, in which $\lambda_b$ is positron annihilation rate in bulk state, $\mu$ is the capture coefficient, $C_d(z)$ is the defect concentration at z depth. The *S* parameter at a certain incident positron energy is actually the weighted superposition of the surface state *S* parameter and each sublayer *S* parameter [32]:

$$S(E) = f_1(E)S_1 + \sum_{i=2}^{m} f_i(E)S_i(E) \tag{5}$$

$$f_1(E) + \sum_{i=2}^{m} f_i(E) = 1 \tag{6}$$

where $S_1$ and $S_i$ are the *S* parameters of the surface state and $i$ th layer, and it is assumed that each sublayer has the same *S* parameter value. $f_1(E)$ and $f_i(E)$ are the positron annihilation probability in the surface state and $i$ th sublayer, respectively.

### 2.4. SRIM Simulation

The theoretical H/He implantation profiles and displacement damage profiles were simulated from SRIM code, taking a displacement threshold energy of approximately 90 eV in CVD-W irradiated by H/He neutral beam. Under the pure H beam irradiation mode, the H ion species acting on the target have a distribution of 22% H of full $E$ (28 keV), 43% of 1/2 $E$ (14 keV) and 35% of 1/3 $E$ [26,34]. In the H + 6 at.% He beam irradiation mode, it was assumed that the H ion species distribution was similar to that of the pure H operation [26,34]. The He ion energy was 28 keV in the H + 6 at.% He operation [26]. The fluence of H and He was $3.4 \times 10^{24}$ m$^{-2}$ and $6.7 \times 10^{24}$ m$^{-2}$ in the simulation, respectively.

## 3. Results and Discussion

### 3.1. SRIM Simulation Results

It can be seen from Figure 1a that the H and He ions are distributed in the depth range of 0–350 nm and 0–230 nm, respectively. The peak of total H ions concentration is ~$1.3 \times 10^5$ at.% at ~60 nm, and it is ~$8.6 \times 10^4$ at.% at ~70 nm for He. As shown in Figure 1b, the peak of displacement damage is ~$1.0 \times 10^3$ dpa at ~30 nm for total H on the left vertical axis and is ~$1.75 \times 10^4$ dpa at ~40 nm for He ions on the right vertical axis. The

relatively large damage values of the two kinds of ions are mainly due to the large fluences of H/He neutral beam.

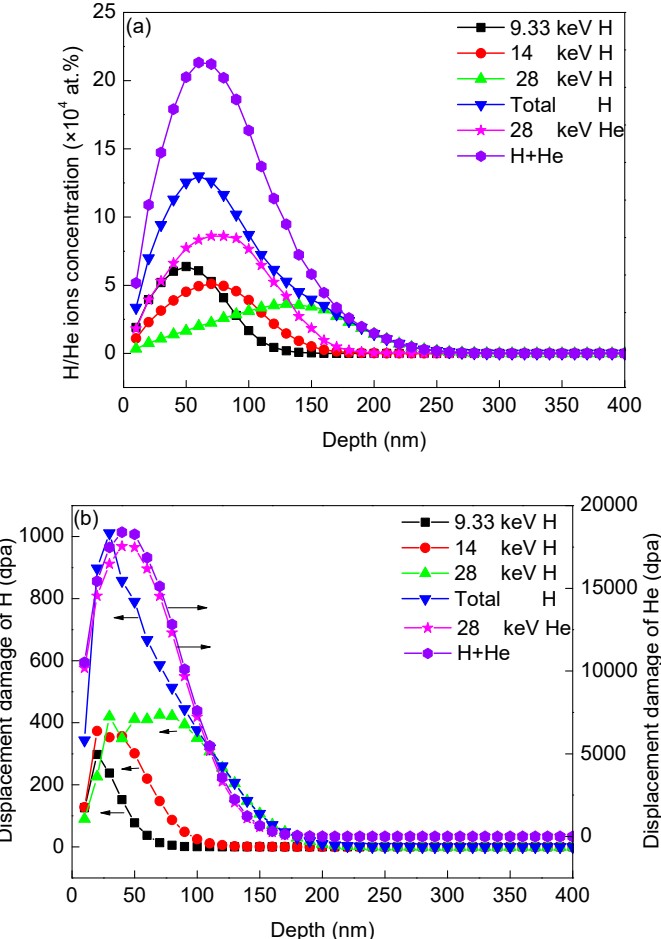

**Figure 1.** SRIM simulation results of (**a**) H and He concentration-depth profiles in CVD-W and (**b**) the corresponding displacement damage-depth profiles.

### 3.2. SEM Results

The morphology of the CVD-W samples after different neutral beam irradiation are shown in Figure 2. As seen in Figure 2a, the surface of unirradiated sample is basically flat with only some scratches formed by the polishing process. The pinhole damage structures can be found on No.1 sample surface (Figure 2b) and most of them disappear on No.2 sample surface (Figure 2c). By comparison of the No.1 and No.2 samples, it can be learned that the surface temperature 1000 °C contributes to the recovery of the irradiation damage in the sample under the pure H beam loading. The surface morphology of the No.3 and No.4 samples after the H + 6 at.% He neutral beam irradiation are presented in Figure 2d,e. The No.3 and No.4 sample surfaces have similar damage structures. It seems that the density of pinhole structures on the No.4 sample surface is higher than that on No.3 sample, indicating that the damage is more serious in the No.4 sample. It is clear that the surface morphology is quite different between the sample surfaces irradiated by the pure H neutral beam (No.1 and No.2) and H + 6 at.% He neutral beam irradiation (No.3 and No.4). The sample surface morphology is closely related to the near surface microstructures formed by sputtering damage in the sample [35]. An experimental study by Liu et al. [26] has shown that the surface morphology of samples would be greatly affected by the mixed He through comparing the sample surfaces irradiated by the pure H neutral beam and H + 6 at.% He neutral beam at the same surface temperature with the same irradiation fluence. It can be speculated that the difference of the surface morphology in the two kinds

of neutral beam irradiation is mainly due to the change of near surface microstructure caused by the introduction of He.

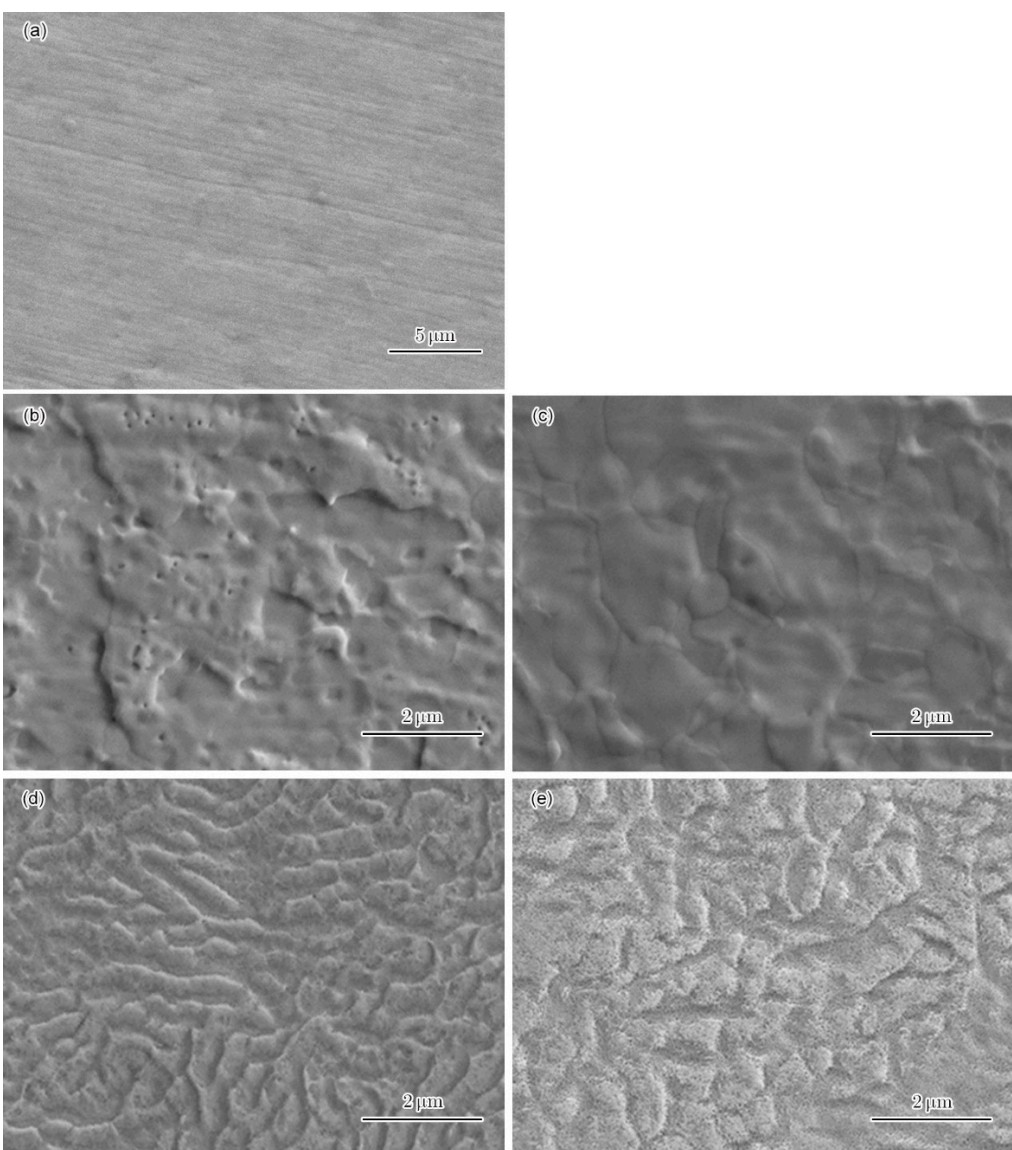

**Figure 2.** The surface morphology of (**a**) the nonirradiated sample and the irradiated samples (**b**) No.1, (**c**) No.2, (**d**) No.3 (**d**), and (**e**) No.4.

### 3.3. DB-SPBA Analysis

Figure 3a shows the *S* parameters as a function of positron beam energy (and the mean positron injection depth shown on the top *X*-axis) of all the samples after different neutral beam irradiation. The scattered dots are the original data, and the different lines are the VEPFIT fitting results. For all the four samples, the *S* parameters first decrease, then increase to a plateau value and finally decrease. The *S* parameters at the positron energy less than 2 keV are relatively large, which would be contributed to two factors. One is that the low-energy positrons have shallow injection depth, the other is that the positrons after thermalizing diffuse to the surface and annihilate with the electrons in the surface [36]. The H or He ions with relative high (dozens of keV) energies irradiated the samples, resulting in the production of a lot of vacancy-type defects, which can be informed from the SRIM simulation. More vacancy-type defects can lead to the larger *S* parameters in the corresponding depth range. With the increase of the positron energy,

the implanted depth becomes deeper and the *S* parameters decrease. The decrease of *S* parameters is due to the decrease of the concentration or the volume of the vacancy-type defects. Figure 3a also shows that *S* parameters of No.1 sample are larger than that of No.2 sample and compared with No.3 sample, the *S* parameters of No.4 sample increase slightly. Both the experimental and theoretical research results [35,37] indicate that He in W has a certain trapping effect on H, hence increasing the retention of H in the samples after the H + 6 at.% He neutral beam irradiation.

The pure H neutral beam irradiated samples contain the injection H atoms and displacement damage, which can be deduced from the SRIM simulation. It is worth noting that the ions injection and damage simulated by SRIM occur at ~−273 °C, which does not consider the temperature effect. Due to the low binding energy of H to vacancy (about 1.28 eV in W) [38], H is easy to be released from the H-V complexes at high temperature of pure H neutral beam irradiation. H can be released from the vacancy or vacancy clusters in W at 227–527 °C [39,40]. Above 527 °C, all the trapped H is most likely to de-trap, migrate to the surface, and then leave the sample, hence the influence of H on the evolution of the defects will no longer be considered [39,40]. In view of this, the effect of H on defect evolution will not be considered in the pure H neutral beam irradiated samples. There are many pinhole-like structures on the surface of No.1 sample and the structures are more likely left after H release. The temperature for monovacancy migration is at ~300 °C to ~500 °C in W [31]. The surface temperature for pure H neutral beam irradiation samples (No.1 and No.2 samples) is higher than 500 °C and the migration and agglomeration of the monovacancy may lead to the formation of the vacancy clusters. The small vacancy clusters are mobile at above 600 °C in W. The migration and agglomeration of the small vacancy clusters are expected to produce large vacancy clusters in the pure H neutral beam irradiation samples.

As for the No.1 sample, the migration and merging of the single vacancies ("Ostwald ripening") and small vacancy clusters in No.1 sample, the defects are likely large volume vacancy clusters [40,41]. The surface temperature of No.2 sample is higher than that of sample No.1. The higher surface temperature is conducive to the recovery of defect damage in No.2 sample. The SEM results above also confirm this point. The recovery of defect damage leads to the decrease of defect concentration in No.2 sample. The higher temperature is also conductive to the mobility and coalescence of vacancies and vacancy clusters in No.2 sample [31]. On the one hand, the migration and combination of the vacancy and vacancy clusters will form larger volume vacancy clusters which cause the increase in S parameters. On the other hand, it will reduce the density of the defects (defect concentration reduction) as indicated by low *S* parameters. It should be kept in mind that the volume increase and concentration reduction of vacancy-type defect are competitive relations. The main effect will dominate the variation trend of *S* parameters. Therefore, the reduction of No.2 sample relative to No.1 sample in *S* parameters is due to the decrease of the vacancy-type defects concentration. The complete recovery of the defects in the H irradiated W is obtained after annealing at about 1527 °C [40]. It can be inferred that part of the vacancy-type defects got recovered in No.2 sample, still leaving a defect damage layer in No.2 sample. To sum up, the larger *S* parameters of No.1 sample are mainly due to the formation of the large size vacancy clusters and the smaller *S* parameters of No.2 sample are mainly attributed to the low concentration of vacancy-type defects.

The H + 6 at.% He neutral beam irradiated samples contain the injection H/He atoms and displacement damage, which can be informed from the SRIM simulation. Because of the retention effect of He on H, the influence of H on vacancy-type defects should be considered in No.3 and No.4 samples. Owing to the relatively high binding energy of He to vacancies (about 4.57 eV in W) [42], the He-V complexes are formed in No.3 and No.4 samples. The H released from H-V complexes can migrate, merge with the existing He-V complexes and form H-He-V complexes in the two samples [43]. The higher irradiated sample surface temperature is more favorable to the migration and combination of the He-V complexes and H-He-V complexes, which leads to the formation of larger He-V complexes

and H-He-V complexes in No.4 sample [31,44,45]. Hence, the increase of the *S* parameters in No.4 sample is caused by the increase of the He related vacancy defects volume.

The *S*-parameters of No.4 sample are smaller than that of No.1 sample in the whole range of the positron injection depth. The fact may be explained by the fill of vacancy clusters with H and He, which promotes the $e^+$ annihilation with H/He $e^-$ that has higher momenta and prevents the $e^+$ annihilation with the low momentum $e^-$ of CVD-W. The main reason why *S* parameters of No.2 sample are smaller than that of No.3 and No.4 samples is that the concentration of the vacancy-type defects is small in No.2 sample.

For CVD-W samples after different neutral beam irradiation, three layers (the surface layer, defect damage layer and deep implanted layer, abbreviated as *S*-layer (1), *S*-layer (2) and *S*-layer (3) respectively) are taken into account in the VEPFIT program to analyze the *S-E* curves. That is, $S(E) = f_1(E)S_1 + \sum_{i=2}^{3} f_i(E)S_i(E)$, $f_1(E) + \sum_{i=2}^{3} f_i(E) = 1$. The depth distributions of the fitting *S* parameters are presented in Figure 3b. The defect damage layer of No.1 sample (about 175 nm) is wider than that in No.2 sample (about 125 nm). It is wider of No.4 sample (about 91 nm) than of No.3 sample (about 45 nm). The surface layers are narrow, with a few nanometers in all samples. The defect damage layer of sample No.1 is the widest in all samples. The upper boundaries of the defect damage layer in No.3 and No.4 are closer to the surface compared to that of No.1 and No.2, which may be attributed to the trapping effect of He on H. According to the previous SRIM simulation in the paper, the He concentration peak position is closer to the surface in comparison with that of H with the same injection energy.

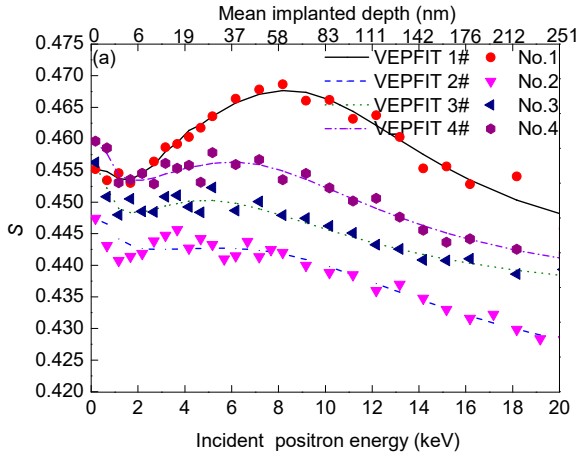

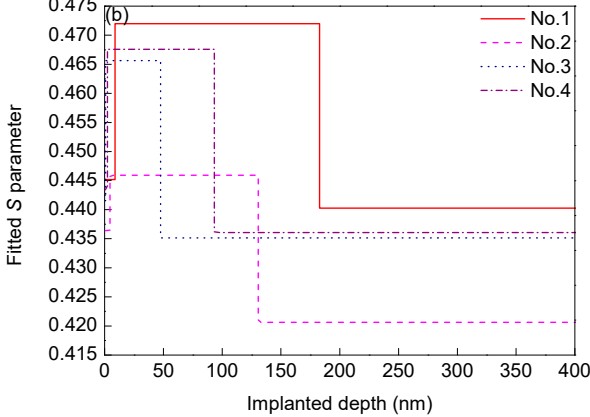

**Figure 3.** *Cont.*

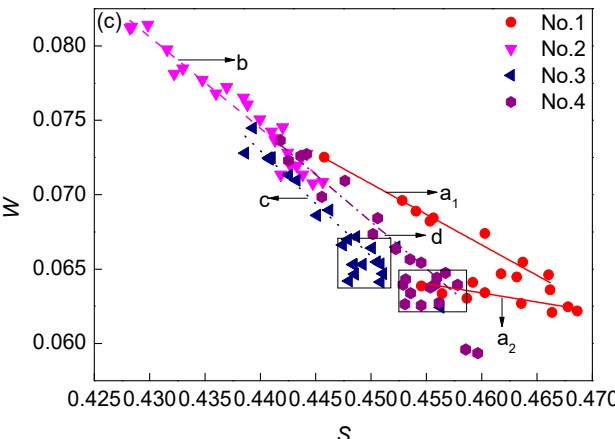

**Figure 3.** (**a**) $S$ parameters as a function of positron beam energy (and the mean positron injection depth shown on the top $X$-axis), the scattered dots are the original data and the different lines are the VEPFIT fitting results, (**b**) the fitted $S$ parameter as a function of implanted depth and (**c**) $W$ as a function of $S$, for CVD-W samples after different neutral beam irradiation conditions.

According to the two-state capture model of defects trapping positrons in the samples, the relationship between $S$ parameters and $W$ parameters is as follows [32]:

$$R = (S_d - S_b)/(W_d - W_b) \qquad (7)$$

where $S_d$ and $S_b$ are the $S$ parameters of the defect and bulk state, $W_d$ and $W_b$ are the $W$ parameters of the defect and bulk state, and R is only related to the defect type. If the measured ($S$, $W$) parameters are distributed on a straight line, revealing the defect types are the same in the sample [46,47]. If the measured ($S$, $W$) parameters are distributed on several different slope lines, then there are several defect types in the sample. The $W$ parameters versus $S$ parameters of CVD-W samples after different neutral beam irradiation are given in Figure 3c. For simplicity, the straight lines of $S$ and $W$ parameters distribution of No.1, No.2, No.3 and No.4 samples are named as lines a, b, c and d, respectively. R = (S_d − S_b)/(W_d − W_b) (6) If the $S$ and $W$ parameters of the same sample are distributed on different straight lines, they are distinguished by the small labels 1 and 2. The $S$ and $W$ parameters of sample No.1 are distributed on two different straight lines (line $a_1$ and line $a_2$), indicating that there are two kinds of vacancy-type defects in the sample. It is speculated that the volumes of vacancy clusters formed in the sample are different, that is, the number of vacancies in vacancy clusters is different [41]. The more vacancies are in vacancy clusters, the larger volume of vacancy clusters is and the greater the slope of the straight line is, as shown in line $a_2$ in the Figure; and vice versa, as shown in line $a_1$ in the Figure.

Obviously, the $W$ ($S$) points are almost in a straight line for sample No.2, which shows that there is only one type of defect in the sample. Moreover, the slope of line b is smaller than that of $a_1$ and $a_2$, indicating that there are less vacancies in vacancy clusters in No.2. When the surface temperature of the irradiated sample is 1000 °C, the defect type is the same under the pure H neutral beam irradiation. Both the $W$ ($S$) parameters of No.3 and No.4 samples are scattered, with the latter being more scattered. This (scatter of $W/S$) may indicate the presence of various types of defects. The slope of line d is slightly larger than that of line c. The probable reason is that the volume of vacancy-type defects in No.4 sample is larger than that in No.3 sample. The defect types are more complicated at 800 °C than that at 700 °C.

By solving the one-dimensional diffusion equation of positron (Equation (3)) and fitting the measured $S$-$E$ curve with Equation (5), the positron diffusion lengths in different layers can be obtained. The positron diffusion lengths in the defect damage layers and deep implanted layers of the samples after different irradiation conditions are shown in

Table 2. Due to the influence of surface effect on the surface layer, the positron diffusion lengths in this layer are not considered here. The positron diffusion lengths of sample No.1 are shorter than that of other samples in the corresponding layers, because the positrons are captured by defects and their diffusion is hindered. According to Equation (4), the positron diffusion length is inversely proportional to the defect concentration. In the layer (2), the small positron diffusion length in the 850 °C sample indicates a large vacancy defect concentration, while the large positron diffusion length in the 1000 °C sample indicates a small vacancy defect concentration. Our fitting results are reasonable and consistent with the above analysis. The difference of positron diffusion lengths between No.3 and No.4 samples is not obvious, owing to the small difference of the vacancy type defects in the two samples.

**Table 2.** The positron diffusion lengths (nm) of CVD-W samples after irradiated with different conditions.

| Sample No. | *S*-layer (2) | *S*-layer (3) |
|------------|---------------|---------------|
| 1 | $19.0 \pm 1.6$ | $30.7 \pm 4.5$ |
| 2 | $41.0 \pm 0.6$ | $101 \pm 11$ |
| 3 | $33 \pm 10$ | $84 \pm 16$ |
| 4 | $37 \pm 9$ | $76 \pm 16$ |

The S parameters are the largest and the defect damage layer is the widest in the sample after the pure H neutral beam irradiation at the surface temperature of 850 °C. The 850 °C sample surface shows larger pinhole structure compared with that of other samples. The samples after the H + 6 at.% He neutral beam irradiation have the similar surface structures, and the difference of vacancy-type defects in size or concentration is small due to the relatively small difference of irradiation condition. The main vacancy-type defects are the vacancy clusters in the samples after the pure H neutral beam irradiation and the He-related vacancy-type defects (He-V complexes and H-He-V complexes) are in the samples after H + 6 at.% He neutral beam irradiation. The different types of defects lead to the different near surface microstructures, which causes different surface morphology between the H + 6 at.% He neutral beam irradiation samples and the pure H neutral beam irradiation samples.

## 4. Conclusions

The evolution of vacancy-type defects in the neutral beam irradiated CVD-W samples was investigated by DB-SPBA and the surface morphology was charactered by SEM. The results can be given as follows.

1.  After the pure H neutral beam irradiation, the *S* parameters in CVD-W sample at the surface temperature of 1000 °C show a decrease compared with that of the 850 °C sample, which could be associated with the reduction in the concentration of vacancy-type defects. The defect damage layer is narrower, and the defect types tend to be the same in the 1000 °C sample. The SEM results demonstrate that the 1000 °C sample, compared with the 850 °C one, exhibits a flatter surface, which indicates that the surface damage was recovered for the 1000 °C sample to some extent.
2.  After the H + 6 at.% He neutral beam irradiation, compared with the CVD-W sample at the surface temperature of 700 °C, sample irradiated at 800 °C showed an increment of the S parameter which is ascribed to an increased volume of vacancy type defects. The defect damage layer is wider in the 800 °C sample. In both the 700 °C sample and 800 °C sample, the defect types are complex. The surface morphology of the 800 °C sample shows more pinhole structures, revealing that the damage is more serious compared with that of the 700 °C sample.

3. The surface morphology of the samples irradiated by the pure H neutral beam is different from that of the samples irradiated by the H + 6 at.% He neutral beam, which is probably caused by the different kinds of micro defects near the surface.

**Author Contributions:** Conceptualization, A.D. and M.G.; methodology, A.D., W.H., X.F. and M.G.; formal analysis, X.T.; investigation, X.T.; data curation, X.T.; writing—original-draft preparation, X.T.; writing—review and editing, X.T., A.D. and M.G. Samples provided, X.L.; Original SEM images provided, X.L. All authors have read and agreed to the published version of the manuscript.

**Funding:** The study is granted by the Nation Natural Science Foundation of China (Grant No.11675114).

**Data Availability Statement:** The data presented in this study is available from the author, X. F. Tian, upon reasonable request.

**Acknowledgments:** We are grateful to Institute of High Energy Physics, Beijing, China for the Doppler broadening spectrum measurements.

**Conflicts of Interest:** The authors declare no conflict of interest.

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
