# Peer review of "Evolution of Defects in CVD-W Irradiated by H/He Neutral Beam Using Positron Annihilation Spectroscopy"

_metals, doi:10.3390/met11020211_

Round 1
Reviewer 1 Report
This paper is sufficiently improved due to the authors effort, as a result, I think that the content of this paper stimulates my intellectual curiosity. However, the ambiguity still remains. Therefore, I judge that this paper should be published after the following revision is carried out.
The (V) of the other points shown in my previous report.
(V)You should explain how to estimate the effective diffuse length in Table 2
You showed line 299-300 of page 9: ‘By solving – can be obtained.’
However, I can not understand why positron diffusion length is related to the measured S-E curve. In the first place, I do not find the measured S-E curve in this paper. Further, I do not understand what the physical meaning of S parameters. Is S parameter the energy distribution for the quantity of positron annihilation or rates f positron annihilation or others?
In addition to this explanation, you should show the procedure how to obtain positron diffusion length.
Reviewer 2 Report
Dear authors,
many thanks for giving a positive feedback to my (and apparently other referees') suggestions. The additional citations are also helpful and correct.
The basics of positron / DBS studies are now well presented.
The SEM micrographs are ok now.
However. some some text passages should be revised for linguistic reasons or scientific clarity:
p. 5 Figure 1: please change 28/3 keV to 9 or 9.3 keV (highl yconfusing to readers)
p. 6 l 198: after [36] the phrase should probably read " the samples are irradiated with H or He ions of relatively high (?) energy (dozens of keV) which rsults in the production of ... defect types as deduced (not informed) from .. the high displacement damage shown by the SRIM simulation Fig 1 ??
Dear authors please clarify the relation of SRIM simulation and defect types.
p 7. l 209 change informed to obtained or deduced (as in line 243)
p. 9 l 295 It is indicated (by what??) the presence the vacancy types present (how) ?
maybe it should read: This (scatter of W/S) may indicated the presence of different / various types of defects (difficult to analyze?).
p. 10 l. 305 change to "... is hindered ..."
section 4. Conclusion
I suggest changing "With" to "After" .. irradiation, assuming that the e+ studies were carried out after the completion of irradion and cooling
2 l. 335 / 336 Pls clarify and "tone down" the deduced relationship of irradiation and S parameter. May be you intended it to read: Samples irradiated at 800 °C showed an increment of the S parameter which is ascribed to an increased volume of vacancy type defects.
Please clarify.
Finally, pls take a short look at the best use of definite and indefinite articles (the / a)
Reviewer 3 Report
The authors have addressed all the comments and the manuscript is much improved with the additional discussions on defect mobiltiies and correlation with their work.
Author Response
Please see the attachment.

This manuscript is a resubmission of an earlier submission. The following is a list of the peer review reports and author responses from that submission.
Round 1
Reviewer 1 Report
There are many ambiguities and no new knowledges which stimulates my intellectual curiosity as seen in Sec.4 Conclusion as follows.
Line 290: I think that the discovery of ‘clusters were formed’ was not obtained from your measurement (Figs. 1-3) because we did not find evidences which show it in results and discussion. In conclusion, you should write only the discovery obtained from your measurement. Lines 293-302: It seems that you mentioned obvious facts which do not stimulate my intellectual curiosity.
Therefore, I do not recommend this paper to be published in Metals.
The other points you should improve
(I)You should explain the principle of measurement of DB-SPBA in Sec.2.
(II)You should show the original measurement data of DB-SPBA in Sec.3.
(III)You should mention the evidences for the sentence of ‘Due to the release of H ---’ (Line 195-197 in page 7) and ‘which leads to decrease of defect concentrate’(Line 200 in page 7).
(IV)Eq.(5) should be removed because there is no quantitative evaluation of R. Further, lines 241-159 after Eq.(5) also be removed because they remain ambiguous.
(V)You should explain how to estimate the effective diffuse length in Table 2
(VI)You should improve the grammar of English.
Reviewer 2 Report
Dear authors,
thanks for submitting this very nice manuscript covering an important detail of fusion reactor development.
The manuscript is well witten with good usage of english and a clear structure.
The abstract and the conclusion are well written and concise
The figure are adequate, well readable and well support the presented findings. However, especially readers of the print version might benefit from selecting SEM micrograph showing a smaller area at higher magnification for Figure 2, to visualize the, e,g., the pinhole structure. The readability might also benefit from refering wore directly to the role of H and He uptake, retention and release on creating these different surface structures. (HR)SEM micrograph of cross sections of the irradiated samples might further elucidate the structural evolution studied, maybe in future work-
The citations are comprehensive, topical and support well the experimental technique, findings and conclusions of that work.
Finally on small point on the experimental section: Did you perform any analysis on the purity and density of the CVD desposited W or do you specify data supplied by the manufacturers ? (please state in the manuscript).
Reviewer 3 Report
Comments in file attached
